# Near-infrared photoimmunotherapy is effective treatment for colorectal cancer in orthotopic nude-mouse models

Hannah M. Hollandsworth[1,2,3], Siamak Amirfakhri[1,2,3], Filemoni Filemoni[1,2,3], Justin Molnar[4], Robert M. Hoffman[1,2,5], Paul Yazaki[4], Michael Bouvet[1,2,3]*

1 Department of Surgery, University of California San Diego, San Diego, CA, United States of America,
2 Moores Cancer Center, University of California San Diego, San Diego, CA, United States of America,
3 Department of Surgery, VA San Diego Healthcare System, San Diego, CA, United States of America,
4 Department of Molecular Imaging and Therapy, Beckman Research Institute City of Hope, Los Angeles, CA, United States of America, 5 AntiCancer Inc., San Diego, CA, United States of America

* mbouvet@ucsd.edu

**Data Availability Statement:** Data is available on the public repository Figshare. https://doi.org/10.6084/m9.figshare.12056157

## Abstract

### Background

Photoimmunotherapy (PIT) employs the use of a near-infrared (NIR) laser to activate an antibody conjugated to a NIR-activatable dye to induce cancer cell death. PIT has shown to be effective in a number of studies, however, there are no data on its use in colorectal cancer in an orthotopic model.

### Methods

Humanized anti-CEA antibody (M5A) was conjugated to NIR-activatable IRDye700DX (M5A-700). PIT was validated in vitro with a colon cancer cell-line, using a laser intensity of either 4 J/cm$^2$, 8 J/cm$^2$, or 16 J/cm$^2$. Orthotopic colon cancer mouse models were established by surgical implantation of LS174T tumor fragments onto the cecum. M5A-700 was administered and PIT was performed 24 hours later using a 690 nm laser. Repeat PIT was performed after 7 days in one group. Control mice received laser treatment only.

### Results

In vitro PIT demonstrated tumor cell death in a laser intensity dose-dependent fashion. In orthotopic models, control mice demonstrated persistent tumor growth. Mice that underwent PIT one time had tumor growth arrested for one week, after which re-growth occurred. The group that received repeated PIT exposure had persistent inhibition of tumor growth.

### Conclusion

PIT arrests tumor growth in colon cancer orthotopic nude-mouse models. Repeated PIT arrests colon cancer growth for a longer period of time. PIT may be a useful therapy in the future as an adjunct to surgical resection or as primary therapy to suppress tumor progression.

**Funding:** The research was supported by VA Merit Review grant numbers 1 I01 BX003856-01A1 and 1 I01 BX004494-01 (MB), and NIH/NCI grant number T32CA121938 (HH). The funder provided support in the form of salaries for authors [HH], but did not have any additional role in the study design, data collection and analysis, decision to publish, or preparation of the manuscript. Dr. Hoffman is affiliated with the commercial company AntiCancer Inc. This commercial affiliation did not provide salaries for any authors and did not have any additional role in the study design, data collection and analysis, decision to publish, or preparation of the manuscript. The specific roles of these authors are articulated in the 'author contributions' section.

**Competing interests:** Drs. Hollandsworth, Bouvet, Amirfakhri and Mr. Filemoni have no competing interests. The authors have read the journal's policy and the following authors have the following competing interests: Drs. Yazaki and Molnar disclose the following information: Patent pending: NIR-conjugated tumor-specific antibodies and uses thereof, Docket 054435-8166. Dr. Hoffman is affiliated with the corporation AntiCancer Inc. This does not alter our adherence to PLOS ONE policies on sharing data and materials.

## Introduction

Photoimmunotherapy (PIT) utilizes a tumor-specific monoclonal antibody conjugated to a photoactivatable dye such as IRDye700DX (IR700, LI-COR, Lincoln, NE) to deliver the photo-active dye to cancer cells [1]. Upon activation of the dye with a near-infrared (NIR) light source, cell membrane damage occurs in cancer cells bound to an antibody against a specific surface antigen of interest [1, 2]. As the dye requires light activation, via laser that emits a similar wavelength, the sequestration of the dye within the tumor causes this treatment to be nontoxic to normal surrounding tissues [3]. Additionally, near-infrared light has been found to be nonionizing and therefore nontoxic to normal tissues that do not have surface bound IR700 [1].

Prior studies of PIT in pancreatic mouse models have targeted tumor-specific surface antigens such as carcinoembryonic antigen [4–6]. A significant decrease in tumor burden was observed in orthotopic pancreatic cancer mouse models that were treated with PIT after administration of a carcinoembryonic antigen (CEA) antibody conjugated to IR700 [4]. Further studies have demonstrated the efficacy of PIT after surgical resection of orthotopic pancreatic cancer mouse models to reduce the rate of recurrence [5, 6].

To date, there are no published data in the literature on the efficacy of PIT in orthotopic models of colorectal cancer. Since targeting the surface antigen CEA has been shown to be effective for PIT in orthotopic pancreatic cancer models, it may also be a useful target for the use of PIT in colorectal cancer as CEA is overexpressed in almost all colorectal cancers [7, 8].

The purpose of the present study is to characterize the efficacy of PIT in orthotopic colorectal cancer mouse models utilizing a humanized anti-CEA monoclonal antibody (m5A) conjugated to a near-infrared fluorophore.

## Materials and methods

### Animals

Athymic nude mice ages 4–6 weeks purchased from Jackson Laboratories (Bar Harbor, ME) were utilized for this study. Mice were maintained in a barrier facility with high-efficiency particulate air filtration and fed an autoclaved laboratory diet. Prior to surgical procedures, mice were anesthetized with an intraperitoneal injection of ketamine and xylazine reconstituted in phosphate-buffered saline (PBS). Immediately after surgical procedures, mice were treated with subcutaneous buprenorphine for pain control. Mice were monitored for five days after procedures for signs of distress or pain, and retreated with buprenorphine when necessary. When the study concluded or if tumor burden became too large, defined as tumor volume > 1500 cm$^3$, mice were euthanized with $CO_2$ inhalation followed by cervical dislocation. This study was carried out in strict accordance with the recommendations in the Guide for the Care and Use of Laboratory Animals of the National Institutes of Health. All animal studies were approved by the San Diego Veterans Administration Medical Center Institutional Animal Care and Use Committee (protocol A17-020).

### Anti-CEA fluorophore conjugation

An Amicon 3 mL stirred cell (Millipore, Burlington, MA) was assembled using a 30 kDa Ultracel Ultrafiltration disc (Millipore, Burlington, MA), placed on a stir table and attached to a flow-through collection reservoir connected to a vacuum pump. One mL of plasma grade water (Fisher Scientific, Waltham, MA) was added to the stirred cell. Fifteen mL of plasma water was added to the supply reservoir and attached to the stirred cell inlet. The plasma water was allowed to flow through the chamber using a light vacuum to maintain a consistent

chamber-fluid level. Once the supply reservoir and chamber were empty, 5 mg (1 ml PBS) of the humanized anti-CEA M5A (M5A) IgG monoclonal antibody (mAb) [9] was added to the chamber. The suspension was dialyzed with 10 diavolumes of basic conjugation buffer. The IRDye-700DX-NHS (LI-COR Biosciences, Lincoln, NE) was dissolved in conjugation buffer to a concentration of 10mg/mL and added at a 10:1 molar ratio. The stirred cell was protected from light and the dye/antibody reaction was stirred for one hour at room temperature. Post conjugation dialysis was performed (1X PBS at pH 7.2). The dialyzed mAb-dye conjugate (1mL) was removed and filtered through a sterile low-protein binding 0.2 µm syringe filter (Pall Corporation, Port Washington, NY) into a sterile 2 mL amber glass vile (Fisher Scientific, Waltham, MA). Protein concentration and degree of labeling were determined using a spectrophotometry at 280 nm and 680 nm as per the dye manufacturer's protocol (LI-COR, Lincoln, NE). Antibody-dye conjugate purity was assessed by a high-performance liquid chromatography size exclusion column (Superdex200) (GE Healthcare Life Sciences, Chicago, IL) monitored at 280 nm and 689 nm.

### In-vitro photoimmunotherapy

Human colon cancer cell line LS174T cells (American Type Culture Collection, Old Town Manassas, VA) were incubated at 37.5 C. We have shown that anti-CEA antibodies target the LS174T human colon cancer cell-line well, which was the basis for choosing this cell line [10]. Cells per well were seeded onto a 96-well plate ($2x10^3$/well) and incubated for 24 hours. Growth medium was then removed and medium-containing M5A-700 was added to all wells, except for four wells that were used as control. The plate was incubated for another 4 hours. Excess medium was removed and replaced with PBS in each well. Increasing intensities of a 690 nm NIR laser (Ultralasers, Inc., Newmarket, Ontario, Canada) were delivered to wells at a distance of 15 cm. Wells were irradiated for 2 minutes total, with laser intensity of either 133 $mW/cm^2$, 266 $mW/cm^2$ or 533 $mW/cm^2$, for a total of 4 $J/cm^2$, 8 $J/cm^2$ or 16 $J/cm^2$, respectively. Cell survival was determined with the Cell Titer 96 AQueous One Solution Cell Proliferation Assay (Promega, Inc. Madison, WI) and cell concentration was measured with a Microplate Reader (Biorad Inc.).

### Tumor establishment

Subcutaneous injection of LS174T cells ($1 x 10^6$) reconstituted in PBS and Matrigel Matrix (Corning, NY) was performed on the bilateral shoulders and flanks of nude mice. Tumors were allowed to grow until 5 mm in diameter. The tumors were then resected and divided into 1 $mm^3$ pieces for orthotopic implantation.

### Orthotopic photoimmunotherapy

In order to establish orthotopic colon cancer models, nude mice (n = 16) were anesthetized as described above. The abdomen was sterilized with 70% ethanol solution. A small midline incision was then made through the skin and abdominal wall muscle. The cecum was carefully removed. A 1 $mm^3$ tumor fragment was implanted onto the serosa of the cecum with an 8–0 nylon suture (Ethicon, Somerville, NJ). The cecum was carefully returned to the peritoneal cavity and the abdominal wall was closed with a 6–0 nylon suture (Ethicon, Somerville, NJ). Tumors were allowed to grow for three weeks [11].

   Mice were randomly divided into three groups; Group 1 control mice (n = 3) were irradiated with the NIR laser without pre-administration of the antibody-fluorophore conjugate; group 2 (n = 6) received PIT only one time; and group 3 (n = 6) received repeated PIT treatment one week after initial treatment. Fig 1 illustrates the experimental protocol. After three

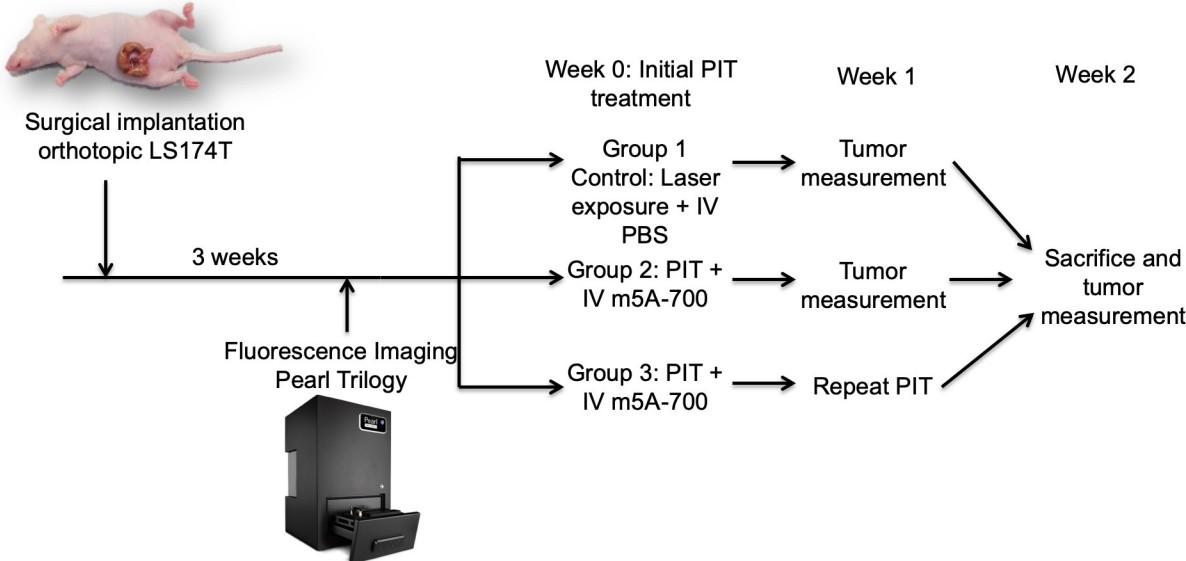

**Fig 1. Experimental protocol.** Surgical implantation of an LS174T tumor fragment (1 mm³) onto the cecum was initially performed. Tumors grew 3 weeks and then were imaged with the Pearl Trilogy Small Animal Imaging System. At week 0, initial PIT treatment was performed. At week 1, group 3 received repeat PIT and tumors were measured with calipers. At week 2, mice were sacrificed and tumor size measured.

weeks of tumor growth, Group 2 and 3 mice were administered 50 μg of m5A-700 reconstituted in PBS via tail vein injection. After 24 hours, mice were anesthetized and tumors were surgically exposed through a midline incision. Tumor size was measured with a caliper. Imaging was obtained on the Pearl Trilogy Small Animal Imaging System prior to treatment. The laser beam was centered on the tumor and treatment was delivered at 150 mW/cm² over 30 minutes for a total delivery of 270 J/cm². Distance from the laser source and the tumor was standardized at 15 cm for each mouse. After treatment, repeat imaging was obtained on the Pearl Trilogy. Group 3 received repeated PIT treatment one week later. Tumors were surgically exposed and measured with calipers weekly for three weeks. Images were analyzed on the Image Studio Small Animal Imaging Analysis Version 5.2 (LI-COR, Lincoln, NE).

## Statistical analysis

Statistical analysis was performed using SPSS Statistics version 24 (IBM, Armonk, NY). Percent cell death was determined for each well for in vitro PIT and mean percent cell death per group was calculated. Imaging analysis was performed using Image Studio Software Small Animal Imaging Analysis (LI-COR, Lincoln, NE). The skin was set as the background and an area of interest around the tumor fluorescence drawn with a minimum of 250 pixels and at least 2.5 standard deviations from the background signal. Maximum tumor signal before and after PIT was determined from the software analysis. Mean tumor fluorescence signal was calculated before and after PIT and means were compared with the univariate student's-t test. Significance was determined with p-value cutoff of 0.05, with 2-tailed analysis. Tumor volume was calculated for each time period using the equation ((width x width x length)/2). Mean and average tumor volume was determined for each group at each tumor measurement time point. The univariate student's-t test was used to compare means between groups at each time point. Significance was determined with p-value cutoff of 0.05, with 2-tailed analysis.

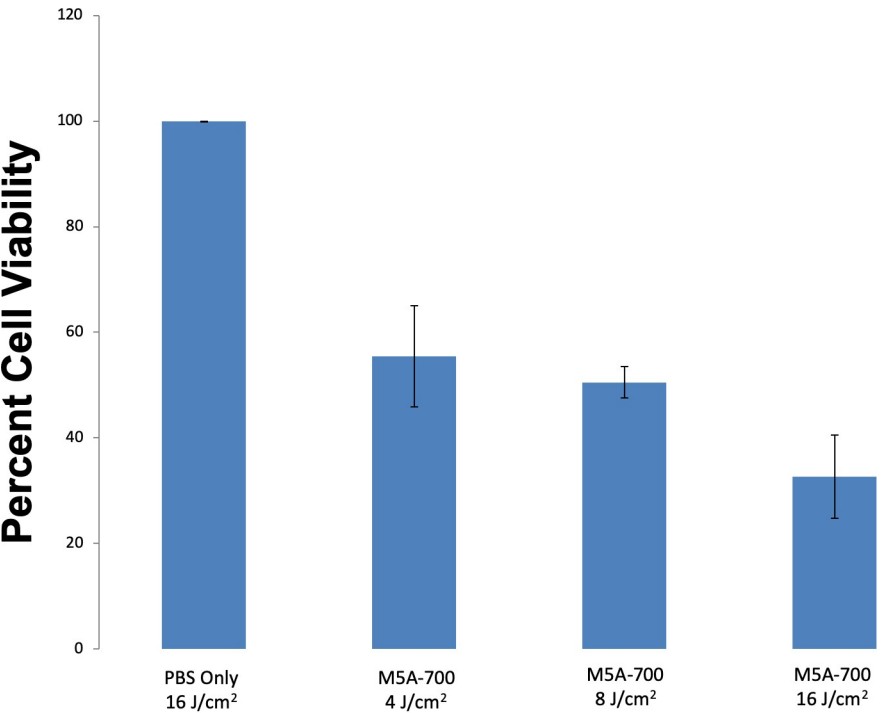

**Fig 2. In vitro PIT cell viability with increasing exposure to laser intensities.** LS174T cells (2,000) were seeded into each well. Treatment groups were incubated with media containing m5A-700. Lower levels of cell viability were demonstrated in a dose-dependent fashion, with the lowest percentage of cell viability in wells that received 16 J/cm$^2$ of laser exposure (p < 0.001).

## Results

### In vitro PIT

After PIT was delivered to cells in vitro, cell survival analysis was performed. Dose response in vitro PIT demonstrated 44.6%, 49.5% and 61.3% cell death with laser intensities of 4 J/cm$^2$, 8 J/cm$^2$ and 16 J/cm$^2$ respectively (p < 0.001, Fig 2).

### Orthotopic PIT

NIR fluorescence signal of the tumor on pre-treatment imaging was compared to signal on post-treatment imaging (n = 12). Prior to treatment, the mean maximum tumor fluorescence was 6.62 (SD ± 3.09). After treatment with PIT, the mean maximum NIR tumor fluorescence decreased to 2.36 (SD ± 0.81), which is indicative of effective PIT treatment (Fig 3). Mean tumor fluorescence before PIT was significantly decreased compared to mean tumor fluorescence after PIT (p < 0.001).

Tumor volumes were measured at weekly intervals with calipers. Fig 4 demonstrates average tumor volume for each group. Control mice demonstrated persistent tumor growth over time. One week after initial PIT treatment, average tumor volume was lower in mice treated with PIT compared to control mice (131.5 mm$^3$ versus 294.1 mm$^3$, respectively, p = 0.063). Mice that received only one PIT treatment did not have a statistically significant difference in tumor volume at week 2 compared to control mice (p = 0.481) Mice that were retreated with

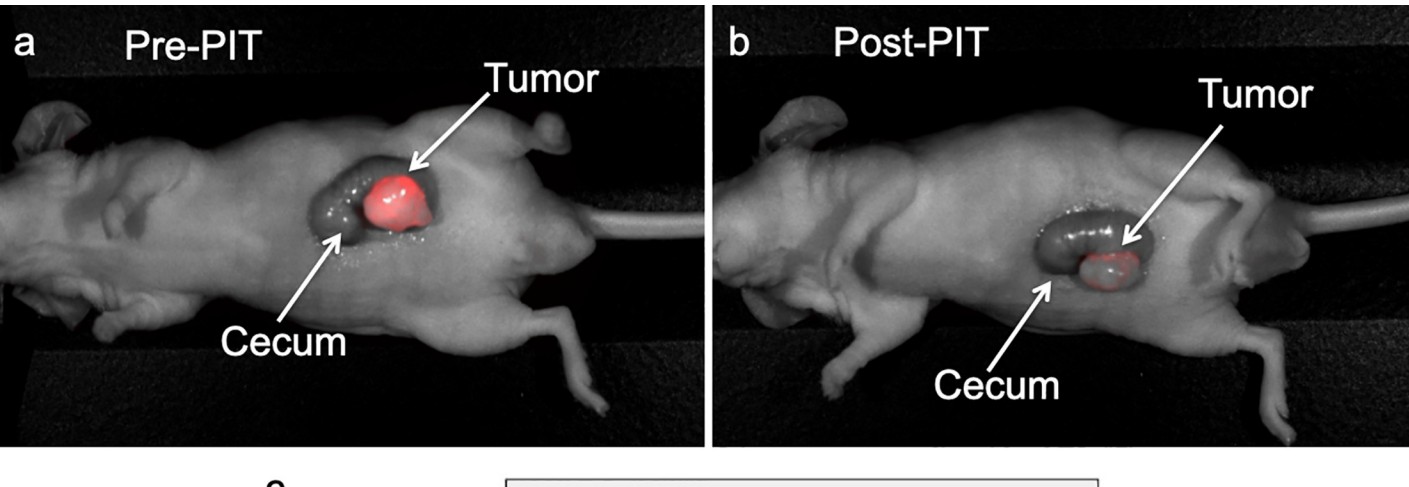

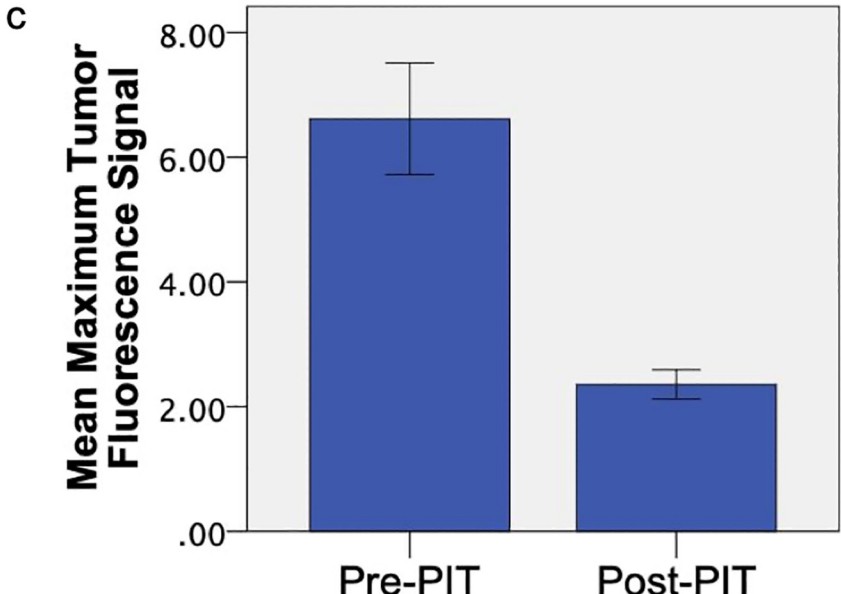

**Fig 3. Comparison of pre-treatment and post-treatment fluorescence imaging in orthotopic LS174T mouse models.** The mouse received 25 µg m5A-700 24 hours before treatment. Prior to PIT treatment (a), tumor margins had a distinct NIR fluorescence signal (maximum tumor fluorescence 4.54). After PIT treatment (b), maximum fluorescence signal decreased to 2.82. (c) Mean maximum fluorescence tumor signal before and after PIT (n = 12). There is a significant difference between mean tumor fluorescence signal before and after treatment with PIT (p < 0.001). Error bars represent standard error of the mean.

PIT one week later had significantly lower tumor volume at week 2 than control mice (p = 0.015).

## Toxicity and adverse effects

Mice were monitored after treatment for adverse effects. During the study period, no mice were observed to have any adverse events. After mice were euthanized, laparotomy was performed to assess intra-abdominal organs. No local toxicity from treatment was detected on any intra-abdominal organs.

## Discussion

In the present study, we initially demonstrated that PIT induces colon cancer cytotoxicity in vitro. This was achieved in a dose-dependent response, which demonstrated higher percentage

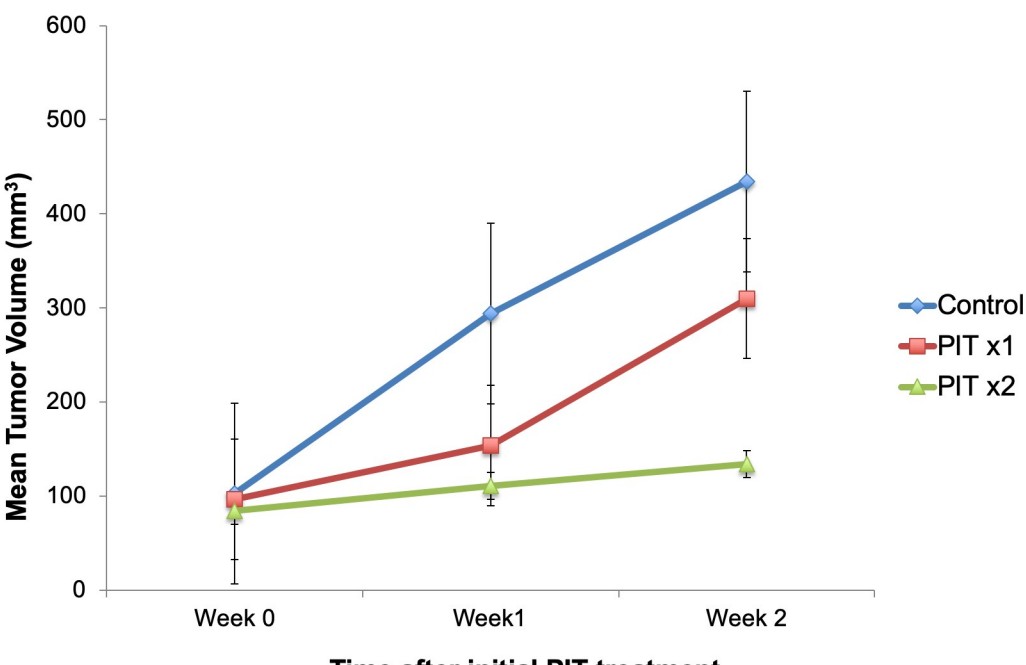

**Fig 4. Tumor volume over time.** PIT was delivered to both treatment groups at week 0. PITx2 group received repeated PIT at week 1. There was a significant difference between PITx2 tumor volume and control tumor volume at week 2 ($p < 0.05$).

of cell death with higher laser intensities. Orthotopic colon cancer mouse models were used to analyze the effects of repeated exposure to PIT therapy. Orthotopic mouse models that received PIT treatment only once had an increased rate of tumor growth one week after treatment with PIT. The increased rate of growth after cessation of PIT treatment compared to the control group may be explained by an increase in local growth factors after treatment. Future studies are needed to test this hypothesis. Mice that were treated with a repeated exposure to PIT one week after initial treatment demonstrated persistently inhibited tumor growth. Tumor volume was significantly different between mice that received multiple PIT treatments and control mice two weeks after treatment.

Ogata et al. assessed repeated exposure to PIT in subcutaneous breast and epithelioid cancer mouse models within 24 hours after initial treatment to determine if there was a greater response to PIT with repeated exposure [12]. The results of Ogata et al demonstrated that PIT causes enhanced permeability of tumor vasculature and increasing the delivery of drug in the first few hours after PIT treatment. Repeated exposure to PIT within the first three hours enhanced cancer cell death [12]. While this is a promising approach for superficial cancer types, this may be difficult to perform clinically for intra-abdominal tumors. Therefore, repeated exposure to PIT seven to ten days after initial treatment may be easier to clinically translate and perform on intra-abdominal tumors to promote enhanced suppression of tumor growth. In clinical applications, the easiest disease process to administer repeated treatments of PIT would be rectal cancer, as this is similar to superficial diseases.

The present study is the first application of PIT in an orthotopic mouse model of colon cancer, which makes the results translatable to the clinic than subcutaneous tumor models. For intra-abdominal colorectal cancers in clinical settings, PIT would have to be administered at the time of a surgical procedure. For colon cancer, this would be most applicable in the setting of non-resectable tumors to decrease tumor burden. In addition to primary therapy, PIT

would likely be effective as a secondary treatment after resection. Although the rate of radial margin positivity, defined as the presence of colon cancer cells at the resected margins, is low after colectomy for colon cancer, the presence of radial margin positivity significantly reduces survival and leads to a higher risk of recurrence [13]. Delivering PIT to the colectomy margins at the time of surgery may decrease the rate of positive radial margins and lead to improved recurrence rates and survival. PIT is delivered using a small catheter that transmits the laser light. Given the small caliber of this catheter, PIT could be administered to patients undergoing a minimally-invasive resection of colorectal cancer.

Another area that PIT may be the most clinically useful is in the treatment of rectal cancer, as mentioned above. It has been previously reported that the rate of recurrence of colorectal cancers is the highest in tumors of rectal origin [14]. This may be explained by the difficulty of obtaining negative margins in the small area afforded by the pelvis. Therefore, it may also prove useful to use PIT as a directed treatment to the resection area in the pelvis after surgical treatment of rectal cancers to reduce or eliminate residual cancer cells. Future mouse model studies with rectal cancer cell lines can provide information on the efficacy of PIT for rectal tumors.

A potential limitation to the use of PIT for colorectal cancer is the ability to administer treatment since colorectal tumors are largely intraluminal. PIT may be most effective for tumor debulking when the tumor has grown out of the serosa or for locally-advanced or metastatic disease. One area that currently has limited treatment guidelines is peritoneal metastases in colorectal cancer. The current guidelines for colorectal peritoneal metastases are unclear; therefore, PIT provides promising treatment for peritoneal metastases that are not amenable to surgical resection. Preclinical studies have demonstrated efficacy in using anti-HER2 monoclonal antibody trastuzumab conjugated to IR700 in the treatment of ovarian cancer peritoneal metastases in mouse models and demonstrated effective cell killing of peritoneal implants [15]. In addition to peritoneal metastases, PIT has been proven to reduce metastatic disease of breast origin to the lungs in preclinical models [16]. These results provide further promise for the potential clinical uses of PIT since the lung is the second most common site of metastases in colorectal cancer [17]. In addition, PIT has the potential to be useful during endoscopic resections of polyps. Prior studies have demonstrated that 51.3% of patients undergoing polypectomy alone had positive resection margins [18]. Performing PIT to the resection bed after polypectomy can be advantageous to treat residual disease and decrease the rate of recurrence and need for further operation.

The mechanism of PIT has been extensively described in prior studies. Kobayashi et al described the leading mechanism as physical changes to the cell membrane that occur after exposure of a tumor with surface-bound monoclonal antibody conjugated to IRDye700DX to a laser light source [1]. After exposure, a "photoinduced ligand reaction" occurs which induces irreversible physical changes to the cell membrane [1]. These changes in the cell membrane allow for rapid entry of water, cellular swelling and eventual bursting of the cell [1, 3]. Activation of the surrounding immune cells and immunogenic cell death may play a role in the tumor suppressive effects of PIT [1, 2]. In the present study, we utilized athymic nude mice that lack T cells, which are not representative of the complex conditions of the immune system in patients. However, even in the absence of T cells, PIT was shown to be effective, suggesting that T cells are not the major factor in immunogenic cell death. Prior to translation into clinical studies, studies will be performed in orthotopic syngeneic models, humanized or genetically engineered mice, all with an intact immune system. Further limitations to the study include the use of a single colon cancer cell line and a short study period, which was due to rapid tumor growth. Future studies with additional cell lines and patient-derived tumor mouse models will increase the translatability and reliability of this technology. This study

serves as a proof-of-principle to demonstrate the effectiveness of PIT in arresting orthotopic colorectal cancer tumor growth, the experimental time was short and detailed histology was not performed. Future studies will include detailed histology to identify characteristics of residual tumor and a longer experimental time prior to translation to clinical studies.

In conclusion, PIT arrested tumor growth in orthotopic colon cancer nude-mouse models. Repeated PIT treatment arrests colon cancer growth for a longer period of time. PIT may be a useful therapy as an adjunct to surgical resection or as primary therapy to suppress tumor progression.

## Author Contributions

**Conceptualization:** Hannah M. Hollandsworth, Siamak Amirfakhri, Justin Molnar, Robert M. Hoffman, Paul Yazaki, Michael Bouvet.

**Data curation:** Hannah M. Hollandsworth, Siamak Amirfakhri, Justin Molnar, Paul Yazaki.

**Formal analysis:** Hannah M. Hollandsworth, Justin Molnar, Paul Yazaki.

**Funding acquisition:** Hannah M. Hollandsworth, Michael Bouvet.

**Investigation:** Hannah M. Hollandsworth, Siamak Amirfakhri, Filemoni Filemoni, Justin Molnar, Michael Bouvet.

**Methodology:** Hannah M. Hollandsworth, Siamak Amirfakhri, Paul Yazaki, Michael Bouvet.

**Project administration:** Hannah M. Hollandsworth, Siamak Amirfakhri, Filemoni Filemoni, Justin Molnar, Paul Yazaki.

**Resources:** Michael Bouvet.

**Supervision:** Robert M. Hoffman, Paul Yazaki, Michael Bouvet.

**Validation:** Hannah M. Hollandsworth.

**Visualization:** Hannah M. Hollandsworth.

**Writing – original draft:** Hannah M. Hollandsworth, Siamak Amirfakhri, Justin Molnar, Paul Yazaki.

**Writing – review & editing:** Hannah M. Hollandsworth, Siamak Amirfakhri, Filemoni Filemoni, Justin Molnar, Robert M. Hoffman, Paul Yazaki, Michael Bouvet.

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
