## [Decision Letter · Decision Letter 0]

19 Mar 2020

PONE-D-20-02950

Near-Infrared Photoimmunotherapy is Effective Treatment for Colorectal Cancer in Orthotopic Nude-Mouse Models

PLOS ONE

Dear Dr Bouvet,

Thank you for submitting your manuscript to PLOS ONE. After careful consideration, we feel that it has merit but does not fully meet PLOS ONE’s publication criteria as it currently stands. Therefore, we invite you to submit a revised version of the manuscript that addresses the points raised during the review process.

Quantification of fluorescent intensity and tumor size need to be provided.  Also histology images should be provided.

What is the rationale for the cell line used? Have experiments been done with other cell lines? What is the rationale for treatment time?

The limitations should be discussed for the mouse model and for technology application.

Are there any considerations for the mechanism(s) involved?

We would appreciate receiving your revised manuscript by May 03 2020 11:59PM. To enhance the reproducibility of your results, we recommend that if applicable you deposit your laboratory protocols in protocols.io, where a protocol can be assigned its own identifier (DOI) such that it can be cited independently in the future. For instructions see: http://journals.plos.org/plosone/s/submission-guidelines#loc-laboratory-protocols

We look forward to receiving your revised manuscript.

Kind regards,

Irina V. Lebedeva, Ph.D.

Academic Editor

PLOS ONE

Journal Requirements:

3. We note that you have a patent relating to material pertinent to this article. Please provide an amended statement of Competing Interests to declare this patent (with details including name and number), along with any other relevant declarations relating to employment, consultancy, patents, products in development or modified products etc. Please confirm that this does not alter your adherence to all PLOS ONE policies on sharing data and materials, as detailed online in our guide for authors http://journals.plos.org/plosone/s/competing-interests by including the following statement: "This does not alter our adherence to  PLOS ONE policies on sharing data and materials.” If there are restrictions on sharing of data and/or materials, please state these. Please note that we cannot proceed with consideration of your article until this information has been declared.

4. Thank you for stating the following in the Financial Disclosure section:

"This study was funded by VA Merit Review grant numbers 1 I01 BX003856-01A1 and 1 I01 BX004494-01 (MB), NIH/NCI T32CA121938 (HH).  "

We note that one or more of the authors are employed by a commercial company: 'AntiCancer Inc'.

5. In the Methods, please provide the formula by which tumour volume was calculated.

Reviewers' comments:

Reviewer's Responses to Questions

**Comments to the Author**

1. Is the manuscript technically sound, and do the data support the conclusions?

Reviewer #1: Partly

Reviewer #2: Yes

2. Has the statistical analysis been performed appropriately and rigorously? 

Reviewer #1: No

Reviewer #2: Yes

3. Have the authors made all data underlying the findings in their manuscript fully available?

Reviewer #1: No

Reviewer #2: Yes

4. Is the manuscript presented in an intelligible fashion and written in standard English?

Reviewer #1: Yes

Reviewer #2: Yes

5. Review Comments to the Author

Reviewer #1: In this manuscript, Hollandsworth et al. describe the application of photoimmunotherapy (PIT) in the treatment of a human colon cancer cell line in vitro and in an orthotopic cecal transplantation model. The authors demonstrate a dose-dependent response to PIT in a cancer cell line. They then show that a single dose of PIT effectively treats the cecal transplantation model. However, a key weakness is that this manuscript fails to provide any mechanism whatsoever to support the conclusion that this technology may be useful for clinical application.

Comments:

1. The authors state that PIT suppressed tumor growth in subcutaneous colon cancer mouse model but this conclusion is demonstrated only in a single mouse image in Figure 3. Quantification of fluorescence intensity and tumor size measurements need to be provided to support this claim. Figure 4 should also be accompanied with quantification and images of tumors.

2. What is the purpose of performing subcutaneous transplant studies if the authors also perform cecal transplants, which are presumably better?

3. Finally, they show inhibition of tumor growth in orthotopic nude mouse models with repeated exposure to PIT one week after initial treatment. Overall the results are promising as this is the first study of PIT in an orthotopic mouse model of colon. However, the conclusions drawn are largely hypothetical as the tumors were grown in athymic nude mice without a competent immune system. Although these mice have dendritic cells, they lack T cells. Thus, this model may have little relevance to humans. Studies in a mouse cell line or organoid transplant model or genetically engineered model would be more relevant.

4. The authors also failed to discuss why the tumors rapidly increased in size following PIT cessation in comparison to control tumors.

5. No experiments were performed to study the possible mechanism of action of PIT in colon cancer. Thus, these experiments add little value to the literature on PIT.

6. Why did the authors select this particular cell line, and why were studies not repeated in additional cell lines or organoid lines?

7. As the authors themselves note, the application of this technology is questionable since the tumor will need to be exposed to the light source. One possible application not discussed by the authors is ablation of positive margins of adenomas during colonoscopy.

8. Why was the treatment only provided for 1-2 weeks? This time scale is extremely short.

9. Histology of the tumors should be provided.

Reviewer #2: The figure legends and results discussion appear to be out of order for several figures. This appears to be an issue of uploading but does not impact the content. The study is technically proficient and highlights an important potential approach to margin positive disease and potentially for peritoneal metastasis.

6. PLOS authors have the option to publish the peer review history of their article (what does this mean?). If published, this will include your full peer review and any attached files.

Reviewer #1: No

Reviewer #2: Yes: Tony R Reid MD, Ph.D.

---

## [Author Response · Author response to Decision Letter 0]

4 Apr 2020

Thank you for your email and reviewers’ comments regarding our manuscript entitled “Near-Infrared Photoimmunotherapy is Effective Treatment for Colorectal Cancer in Orthotopic Nude-Mouse Models.” Thank you also for your interest and the thoughtful comments provided by the Reviewers. For our revision submission, we are providing a revised manuscript that addresses all Reviewer comments and suggestions.

Below, please find the Reviewer’s comments and how each was addressed in the revised manuscript. The Reviewer comments are numbered and our responses to each comment are below in bold.

 We thank you again for the opportunity to submit this revised manuscript and look forward to your comments.

Reviewers' comments:

Reviewer 1: 

1. The authors state that PIT suppressed tumor growth in subcutaneous colon cancer mouse model but this conclusion is demonstrated only in a single mouse image in Figure 3. Quantification of fluorescence intensity and tumor size measurements need to be provided to support this claim. Figure 4 should also be accompanied with quantification and images of tumors.

Figure 4 legend (now labelled Figure 3) and the Results now include quantification of fluorescence intensity in the orthotopic models. 

Please see answer to question 2 below. 

2. What is the purpose of performing subcutaneous transplant studies if the authors also perform cecal transplants, which are presumably better?

Data on subcutaneous models was removed from the revised manuscript.

3. Finally, they show inhibition of tumor growth in orthotopic nude mouse models with repeated exposure to PIT one week after initial treatment. Overall the results are promising as this is the first study of PIT in an orthotopic mouse model of colon. However, the conclusions drawn are largely hypothetical as the tumors were grown in athymic nude mice without a competent immune system. Although these mice have dendritic cells, they lack T cells. Thus, this model may have little relevance to humans. Studies in a mouse cell line or organoid transplant model or genetically engineered model would be more relevant.

A section in the discussion has been included to discuss the limitation highlighted above: “In the present study, we utilized athymic nude mice that lack T cells, which are not representative of the complex conditions of the immune system in patients. However, even in the absence of T cells, PIT was shown to be effective, suggesting that T cells are not the major factor in immunogenic cell death. Prior to translation into clinical studies, we will perform PIT on orthotopic syngeneic models, humanized or genetically engineered mice, all with an intact immune system.” 

4. The authors also failed to discuss why the tumors rapidly increased in size following PIT cessation in comparison to control tumors.

This is an important question. We can only speculate that the first PIT caused the production of growth factors, such is sometimes the case in surgery, where residual tumors grow faster. Future experiments are needed to answer this question. This is stated in the revised manuscript. 

5. No experiments were performed to study the possible mechanism of action of PIT in colon cancer. Thus, these experiments add little value to the literature on PIT.

The present studies are a proof-of-principle that PIT is effective on orthotopic models of colon cancer, as previous studies were only on subcutaneous colon cancer models, which are artificial. This is stated in the revised manuscript. However, general mechanisms of PIT are discussed in the revised manuscript. 

6. Why did the authors select this particular cell line, and why were studies not repeated in additional cell lines or organoid lines?

We have done previous studies showing human colon cancer LS174T cell line is targeted well by anti-CEA antibodies (1) and would be appropriate for an orthotopic PIT study and proof-of-principle. This is stated in the revised manuscript. 

7. As the authors themselves note, the application of this technology is questionable since the tumor will need to be exposed to the light source. One possible application not discussed by the authors is ablation of positive margins of adenomas during colonoscopy.

PIT can be an intra-operative procedure; therefore, illuminating an intra-abdominal tumor is feasible. This is discussed in the revised version. The potential application for polyp margins during endoscopy is also now included in the Discussion. 

8. Why was the treatment only provided for 1-2 weeks? This time scale is extremely short.

The limitation of the time scale is included in the Discussion. This also includes the reason for short time period, which was tumor growth in the control group that exceeded the allowed tumor burden. It was discussed that survival and recurrence studies can increase clinical applicability of PIT in colorectal cancer. 

9. Histology of the tumors should be provided.

As the present study was a proof-of-principle study that PIT could arrest an orthotopic tumor, histology studies were not done. Detailed histology studies will be done in the future. This is discussed in the revised edition. 

As part of this submission, we have included a revised version of the main manuscript file that shows changes made by highlighting, as well as a clean version of the revised manuscript. Again, we thank the editors and reviewers for their efforts in improving our manuscript. We would be happy to answer any further questions and/or concerns.

---

## [Decision Letter · Decision Letter 1]

5 May 2020

PONE-D-20-02950R1

Near-Infrared Photoimmunotherapy is Effective Treatment for Colorectal Cancer in Orthotopic Nude-Mouse Models

PLOS ONE

Dear Dr Bouvet,

Thank you for submitting your manuscript to PLOS ONE. After careful consideration, we feel that it has merit but does not fully meet PLOS ONE’s publication criteria as it currently stands. Therefore, we invite you to submit a revised version of the manuscript that addresses the points raised during the review process.

We would appreciate receiving your revised manuscript by Jun 19 2020 11:59PM. To enhance the reproducibility of your results, we recommend that if applicable you deposit your laboratory protocols in protocols.io, where a protocol can be assigned its own identifier (DOI) such that it can be cited independently in the future. For instructions see: http://journals.plos.org/plosone/s/submission-guidelines#loc-laboratory-protocols

We look forward to receiving your revised manuscript.

Kind regards,

Irina V. Lebedeva, Ph.D.

Academic Editor

PLOS ONE

Reviewers' comments:

Reviewer's Responses to Questions

**Comments to the Author**

1. If the authors have adequately addressed your comments raised in a previous round of review and you feel that this manuscript is now acceptable for publication, you may indicate that here to bypass the “Comments to the Author” section, enter your conflict of interest statement in the “Confidential to Editor” section, and submit your "Accept" recommendation.

Reviewer #1: All comments have been addressed

Reviewer #2: All comments have been addressed

2. Is the manuscript technically sound, and do the data support the conclusions?

Reviewer #1: Yes

Reviewer #2: Yes

3. Has the statistical analysis been performed appropriately and rigorously? 

Reviewer #1: Yes

Reviewer #2: Yes

4. Have the authors made all data underlying the findings in their manuscript fully available?

Reviewer #1: Yes

Reviewer #2: Yes

5. Is the manuscript presented in an intelligible fashion and written in standard English?

Reviewer #1: Yes

Reviewer #2: Yes

6. Review Comments to the Author

Reviewer #1: The author's has appropriately limited the scope of their conclusions and added appropriate discussion. Below please see our comments about specific points with minor revision requests.

Reviewers' Response to comments:

1. The authors state that PIT suppressed tumor growth in subcutaneous colon cancer

mouse model but this conclusion is demonstrated only in a single mouse image in

Figure 3. Quantification of fluorescence intensity and tumor size measurements need

to be provided to support this claim. Figure 4 should also be accompanied with

quantification and images of tumors.

Author’s Response: Figure 4 legend (now labelled Figure 3) and the Results now include quantification of fluorescence intensity in the orthotopic models.

Please see answer to question 2 below.

Reviewer’s Response:

How many fluorescence measurements were calculated? It would strengthen the claim if this data was shared as a graph in updated Figure 3.

2. What is the purpose of performing subcutaneous transplant studies if the authors

also perform cecal transplants, which are presumably better?

Author’s Response: Data on subcutaneous models was removed from the revised manuscript.

Reviewer’s Response:

Author’s can include other relevant experiments as supplemental figures.

3. Finally, they show inhibition of tumor growth in orthotopic nude mouse models with

repeated exposure to PIT one week after initial treatment. Overall the results are

promising as this is the first study of PIT in an orthotopic mouse model of colon.

However, the conclusions drawn are largely hypothetical as the tumors were grown in

athymic nude mice without a competent immune system. Although these mice have

dendritic cells, they lack T cells. Thus, this model may have little relevance to humans.

Studies in a mouse cell line or organoid transplant model or genetically engineered

model would be more relevant.

Author’s Response: A section in the discussion has been included to discuss the limitation highlighted above: “In the present study, we utilized athymic nude mice that lack T cells, which are not representative of the complex conditions of the immune system in patients. However, even in the absence of T cells, PIT was shown to be effective, suggesting that T cells are not the major factor in immunogenic cell death. Prior to translation into clinical studies, we will perform PIT on orthotopic syngeneic models, humanized or genetically engineered mice, all with an intact immune system.”

Reviewer’s Response:

This is acceptable.

4. The authors also failed to discuss why the tumors rapidly increased in size following

PIT cessation in comparison to control tumors.

Author’s Response: This is an important question. We can only speculate that the first PIT caused the production of growth factors, such is sometimes the case in surgery, where residual tumors grow faster. Future experiments are needed to answer this question. This is

stated in the revised manuscript.

Reviewer’s Response:

Histology of the tumors on future studies can also help identify characteristics of the residual tumor.

5. No experiments were performed to study the possible mechanism of action of PIT in

colon cancer. Thus, these experiments add little value to the literature on PIT.

Author’s Response: The present studies are a proof-of-principle that PIT is effective on orthotopic models of colon cancer, as previous studies were only on subcutaneous colon cancer models, which are artificial. This is stated in the revised manuscript. However, general mechanisms of PIT are discussed in the revised manuscript.

Reviewer’s Response:

The new information provided will help readers understand the possible future directions.

6. Why did the authors select this particular cell line, and why were studies not

repeated in additional cell lines or organoid lines?

Author’s Response:

We have done previous studies showing human colon cancer LS174T cell line is

targeted well by anti-CEA antibodies (1) and would be appropriate for an orthotopic PIT

study and proof-of-principle. This is stated in the revised manuscript.

7. As the authors themselves note, the application of this technology is questionable

since the tumor will need to be exposed to the light source. One possible application

not discussed by the authors is ablation of positive margins of adenomas during

colonoscopy.

Author’s Response:

PIT can be an intra-operative procedure; therefore, illuminating an intra-abdominal

tumor is feasible. This is discussed in the revised version. The potential application for

polyp margins during endoscopy is also now included in the Discussion.

8. Why was the treatment only provided for 1-2 weeks? This time scale is extremely

short.

Author’s Response:

The limitation of the time scale is included in the Discussion. This also includes the

reason for short time period, which was tumor growth in the control group that

exceeded the allowed tumor burden. It was discussed that survival and recurrence

studies can increase clinical applicability of PIT in colorectal cancer.

9. Histology of the tumors should be provided.

Author’s Response:

As the present study was a proof-of-principle study that PIT could arrest an orthotopic

tumor, histology studies were not done. Detailed histology studies will be done in the

future. This is discussed in the revised edition.

As part of this submission, we have included a revised version of the main manuscript

file that shows changes made by highlighting, as well as a clean version of the revised

manuscript. Again, we thank the editors and reviewers for their efforts in improving our

manuscript. We would be happy to answer any further questions and/or concerns.

Reviewer #2: The study demonstrates dose dependent tumor cell killing with PIT and the potential application of this technology to surgical settings. The non-toxic aspect of PIT in the absence of the conjugated antibody offers the potential to treat otherwise surgically inoperable disease as well as debulk disease. The paper offers an important potential path to managing complex intra-abdominal metastatic disease.

7. PLOS authors have the option to publish the peer review history of their article (what does this mean?). If published, this will include your full peer review and any attached files.

Reviewer #1: No

Reviewer #2: Yes: Tony Reid MD, Ph.D.

---

## [Author Response · Author response to Decision Letter 1]

8 May 2020

May 7, 2020

To: Drs. Joreg Heber and Irina V. Lebedeva, Editor-in Chief and Academic Editor, PLOS One

Manuscript – “Near-Infrared Photoimmunotherapy is Effective Treatment for Colorectal Cancer in Orthotopic Nude-Mouse Models”

Drs. Heber and Lebedeva,

Thank you for your email and reviewers’ comments regarding our manuscript entitled “Near-Infrared Photoimmunotherapy is Effective Treatment for Colorectal Cancer in Orthotopic Nude-Mouse Models.” Thank you also for your interest and the thoughtful comments provided by the Reviewers. For our revision submission, we are providing a revised manuscript that addresses all Reviewer comments and suggestions.

Below, please find the Reviewer’s comments and how each was addressed in the revised manuscript. The most recent Reviewer comments are italicized and our responses to each comment are below in bold. 

 We thank you again for the opportunity to submit this revised manuscript and look forward to your comments.

Reviewers' Response to comments:

1. The authors state that PIT suppressed tumor growth in subcutaneous colon cancer

mouse model but this conclusion is demonstrated only in a single mouse image in

Figure 3. Quantification of fluorescence intensity and tumor size measurements need

to be provided to support this claim. Figure 4 should also be accompanied with

quantification and images of tumors.

Author’s Response: Figure 4 legend (now labelled Figure 3) and the Results now include quantification of fluorescence intensity in the orthotopic models.

Please see answer to question 2 below.

Reviewer’s Response:

How many fluorescence measurements were calculated? It would strengthen the claim if this data was shared as a graph in updated Figure 3.

Figure 3 has been updated to include a graph depicting the difference in mean maximum tumor fluorescence signal before and after PIT. The figure legend provides description of how many fluorescence measurements were calculated and statistical significance. Materials and methods have been revised to include image analysis and statistical analysis of tumor fluorescence. 

2. What is the purpose of performing subcutaneous transplant studies if the authors

also perform cecal transplants, which are presumably better?

Author’s Response: Data on subcutaneous models was removed from the revised manuscript.

Reviewer’s Response:

Author’s can include other relevant experiments as supplemental figures.

All relevant experiments are presented in the main manuscript. 

3. Finally, they show inhibition of tumor growth in orthotopic nude mouse models with

repeated exposure to PIT one week after initial treatment. Overall the results are

promising as this is the first study of PIT in an orthotopic mouse model of colon.

However, the conclusions drawn are largely hypothetical as the tumors were grown in

athymic nude mice without a competent immune system. Although these mice have

dendritic cells, they lack T cells. Thus, this model may have little relevance to humans.

Studies in a mouse cell line or organoid transplant model or genetically engineered

model would be more relevant.

Author’s Response: A section in the discussion has been included to discuss the limitation highlighted above: “In the present study, we utilized athymic nude mice that lack T cells, which are not representative of the complex conditions of the immune system in patients. However, even in the absence of T cells, PIT was shown to be effective, suggesting that T cells are not the major factor in immunogenic cell death. Prior to translation into clinical studies, we will perform PIT on orthotopic syngeneic models, humanized or genetically engineered mice, all with an intact immune system.”

Reviewer’s Response:

This is acceptable.

4. The authors also failed to discuss why the tumors rapidly increased in size following

PIT cessation in comparison to control tumors.

Author’s Response: This is an important question. We can only speculate that the first PIT caused the production of growth factors, such is sometimes the case in surgery, where residual tumors grow faster. Future experiments are needed to answer this question. This is

stated in the revised manuscript.

Reviewer’s Response:

Histology of the tumors on future studies can also help identify characteristics of the residual tumor.

This point was added to the revised discussion. 

5. No experiments were performed to study the possible mechanism of action of PIT in

colon cancer. Thus, these experiments add little value to the literature on PIT.

Author’s Response: The present studies are a proof-of-principle that PIT is effective on orthotopic models of colon cancer, as previous studies were only on subcutaneous colon cancer models, which are artificial. This is stated in the revised manuscript. However, general mechanisms of PIT are discussed in the revised manuscript.

Reviewer’s Response:

The new information provided will help readers understand the possible future directions.

6. Why did the authors select this particular cell line, and why were studies not

repeated in additional cell lines or organoid lines?

Author’s Response:

We have done previous studies showing human colon cancer LS174T cell line is

targeted well by anti-CEA antibodies (1) and would be appropriate for an orthotopic PIT

study and proof-of-principle. This is stated in the revised manuscript.

7. As the authors themselves note, the application of this technology is questionable

since the tumor will need to be exposed to the light source. One possible application

not discussed by the authors is ablation of positive margins of adenomas during

colonoscopy.

Author’s Response:

PIT can be an intra-operative procedure; therefore, illuminating an intra-abdominal

tumor is feasible. This is discussed in the revised version. The potential application for

polyp margins during endoscopy is also now included in the Discussion.

8. Why was the treatment only provided for 1-2 weeks? This time scale is extremely

short.

Author’s Response:

The limitation of the time scale is included in the Discussion. This also includes the

reason for short time period, which was tumor growth in the control group that

exceeded the allowed tumor burden. It was discussed that survival and recurrence

studies can increase clinical applicability of PIT in colorectal cancer.

9. Histology of the tumors should be provided.

Author’s Response:

As the present study was a proof-of-principle study that PIT could arrest an orthotopic

tumor, histology studies were not done. Detailed histology studies will be done in the

future. This is discussed in the revised edition.

As part of this submission, we have included a revised version of the main manuscript

file that shows changes made by highlighting, as well as a clean version of the revised

manuscript. Again, we thank the editors and reviewers for their efforts in improving our

manuscript. We would be happy to answer any further questions and/or concerns.

As part of this submission, we have included a revised version of the main manuscript file that shows changes made by highlighting, as well as a clean version of the revised manuscript. Again, we thank the editors and reviewers for their efforts in improving our manuscript. We would be happy to answer any further questions and/or concerns.

Sincerely,

Hannah Hollandsworth, MD

---

## [Decision Letter · Decision Letter 2]

1 Jun 2020

Near-Infrared Photoimmunotherapy is Effective Treatment for Colorectal Cancer in Orthotopic Nude-Mouse Models

PONE-D-20-02950R2

Dear Dr. Bouvet,

We are pleased to inform you that your manuscript has been judged scientifically suitable for publication and will be formally accepted for publication once it complies with all outstanding technical requirements.

With kind regards,

Irina V. Lebedeva, Ph.D.

Academic Editor

PLOS ONE

Additional Editor Comments (optional):

Reviewers' comments:

Reviewer's Responses to Questions

**Comments to the Author**

1. If the authors have adequately addressed your comments raised in a previous round of review and you feel that this manuscript is now acceptable for publication, you may indicate that here to bypass the “Comments to the Author” section, enter your conflict of interest statement in the “Confidential to Editor” section, and submit your "Accept" recommendation.

Reviewer #1: All comments have been addressed

2. Is the manuscript technically sound, and do the data support the conclusions?

Reviewer #1: Yes

3. Has the statistical analysis been performed appropriately and rigorously? 

Reviewer #1: Yes

4. Have the authors made all data underlying the findings in their manuscript fully available?

Reviewer #1: Yes

5. Is the manuscript presented in an intelligible fashion and written in standard English?

Reviewer #1: Yes

6. Review Comments to the Author

Reviewer #1: All review questions / comments have been addressed. I recommend proceeding with acceptance and publication.

7. PLOS authors have the option to publish the peer review history of their article (what does this mean?). If published, this will include your full peer review and any attached files.

Reviewer #1: Yes: Jatin Roper

---

## [Editor Report · Acceptance letter]

3 Jun 2020

PONE-D-20-02950R2 

Near-Infrared Photoimmunotherapy is Effective Treatment for Colorectal Cancer in Orthotopic Nude-Mouse Models 

Dear Dr. Bouvet:

I'm pleased to inform you that your manuscript has been deemed suitable for publication in PLOS ONE. Congratulations! Your manuscript is now with our production department. 

Kind regards, 

on behalf of

Dr. Irina V. Lebedeva 

Academic Editor

PLOS ONE